# Real-time photonic blind interference cancellation

Joshua C. Lederman [1] ✉, Weipeng Zhang [1], Thomas Ferreira de Lima[1,2], Eric C. Blow[1,2], Simon Bilodeau[1], Bhavin J. Shastri [3] & Paul R. Prucnal [1]

mmWave devices can broadcast multiple spatially-separated data streams simultaneously in order to increase data transfer rates. Data transfer can, however, be compromised by interference. Photonic blind interference cancellation systems offer a power-efficient means of mitigating interference, but previous demonstrations of such systems have been limited by high latencies and the need for regular calibration. Here, we demonstrate real-time photonic blind interference cancellation using an FPGA-photonic system executing a zero-calibration control algorithm. Our system offers a greater than 200-fold reduction in latency compared to previous work, enabling sub-second cancellation weight identification. We further investigate key trade-offs between system latency, power consumption, and success rate, and we validate sub-Nyquist sampling for blind interference cancellation. We estimate that photonic interference cancellation can reduce the power required for digitization and signal recovery by greater than 74 times compared to the digital electronic alternative.

Technological development has driven an ever-increasing demand for wireless communication bandwidth[1–3]. With sub-7 GHz bands heavily utilized, the industry has turned to 30–300 GHz (mmWave) bands to meet these growing needs[4,5]. At these frequencies, further capacity gains can be realized using beamforming—the angular steering and filtering of radio-frequency (RF) signals to optimize wireless communication links. mmWave beamforming may be implemented using an array of mm-scale antennas in one package[3–6]. Hybrid analog-digital beamformers are capable of transmitting or receiving signals at multiple angles simultaneously, allowing independent data streams to be transmitted along different spatial paths concurrently (Fig. 1)[3–5,7,8]. This multiple-input multiple-output (MIMO) approach multiplies the capacity of an RF link[3–6].

Beamforming receivers are imperfect, receiving signals from off-target angles, particularly those close to the angles of target signals. Interfering signals arriving at such angles introduce noise that degrades network capacity[5,9–12]. This interference may stem from other spatial channels, including those of other devices on the network sharing the same spectral resources as in a multi-user MIMO system (Fig. 1a)[4,5], or from a malicious actor (Fig. 1b).

Interference may be mitigated by isolating each incoming signal according to its angle of incidence and subtracting it from the target signals, improving their signal-to-noise ratios (SNRs)[5,9,13–15]. When implemented digitally, this technique requires a power-hungry RF chain for every interference source in order to generate the associated digital reference signal[4,7,12,16]. Analog interference cancellation, where interference is subtracted prior to signal digitization, can reduce system power consumption. We propose to implement it using RF photonics.

The RF photonics platform, where RF signals are modulated onto optical carriers and processed in the optical domain, offers high bandwidths, low loss, and resistance to electro-magnetic interference[17]. A single integrated silicon photonic waveguide can carry dozens of high-bandwidth RF signals each on a separate wavelength, enabling hardware-efficient interconnection[18]. These wavelength-division-multiplexed (WDM) RF signals can be weighted and summed in parallel using micro-ring resonators (MRRs) and

[1]Department of Electrical and Computer Engineering, Princeton University, Princeton, NJ 08544, USA. [2]NEC Laboratories America, Princeton, NJ 08540, USA. [3]Department of Physics, Engineering Physics & Astronomy, Queen's University, Kingston, ON K7L 3N6, Canada. ✉e-mail: jlederman@princeton.edu

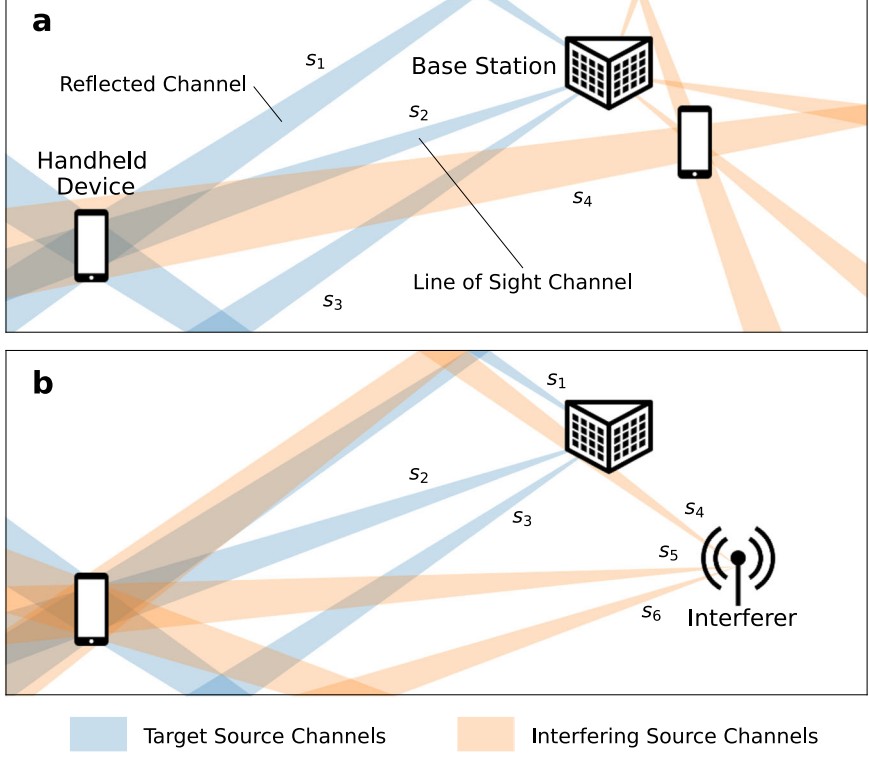

**Fig. 1 | mmWave interference scenarios. a** A spatial channel associated with one device interferes with another device's spatial channels. **b** A malicious source directs interference at a device. Target channels are shown in blue and interfering channels in orange.

photodetectors, implementing tunable linear signal combination[19]. Applications of RF-photonic linear combination include self-interference cancellation[20,21] and blind interference cancellation[22–27]. Notably, we have recently demonstrated photonic blind interference cancellation (PBIC) with 9 bits of weight precision and effective control of signals from DC to 19.2 GHz[27]. However, like all previous MRR photonic systems, that PBIC system required re-calibration after any small shift in operating temperature or optical input power, and identifying the correct cancellation weights required minutes, incompatible with real-time operation.

In this work, we demonstrate a PBIC system with key advancements that address limitations observed in prior work. Our system relies on a zero-calibration approach to MRR control for PBIC that greatly reduces the complexity of adapting to changes in temperature and optical power. Digital processing in our system is implemented on a Xilinx Zynq chip containing a field-programmable gate array (FPGA) and a central processing unit (CPU), resulting in a greater than 200-fold reduction in latency and sub-second cancellation weight identification. We demonstrate low-latency coordinated processing between an MRR photonic system and an FPGA as well as real-time applied photonic weight adaptation. Our results highlight that the statistic sampling rate and sample count represent critical parameters impacting the latency, power consumption, and success rate of a PBIC system. We establish that sub-Nyquist sampling is a crucial technique for reducing power consumption without compromising PBIC success. Finally, we propose a mmWave beamforming receiver architecture capable of PBIC and estimate that it can achieve a greater than 74-fold reduction in digitization and signal recovery power in comparison to the conventional digital electronic alternative.

## Results

We consider a set of $N_s$ spatially-separated transmitters communicating with a single receiver. There are $N_s$ independent source signals $\mathbf{s}(t) = \{s_1(t), s_2(t), \ldots, s_{N_s}(t)\}$. These source signals mix over the air, such

that the receiver detects a set of $N_r$ distinct linear mixtures of those signals $\mathbf{r}(t) = \{r_1(t), r_2(t), \ldots, r_{N_r}(t)\}$, where we require $N_r \geq N_s$. The mixing process, neglecting non-interference noise sources, may be modeled:

$$\mathbf{r}(t) = \mathbf{M}\mathbf{s}(t) \tag{1}$$

where $\mathbf{M} \in \mathbb{R}^{N_r \times N_s}$ represents an unknown mixing matrix.

Blind interference cancellation describes the task of recovering, from the received signals, a subsection of the source signals numbering $N_t$, the target source signals, where $N_t \leq N_s$. For each of these target source signal $s_i(t)$, there exists a cancellation weight vector $\mathbf{c}_i$ that recovers the source:

$$s_i(t) = \mathbf{c}_i \cdot \mathbf{r}(t) = \mathbf{c}_i^T \mathbf{M}\mathbf{s}(t) \tag{2}$$

Under photonic blind interference cancellation, also called photonic blind source separation, this signal recovery is implemented in the analog domain with photonics[22–27]. As shown in Fig. 2, each received signal is modulated onto a distinct wavelength of light. The signals are multiplexed and coupled onto a silicon photonic chip. On-chip tunable micro-ring resonators (MRRs) apply photonic weights $\mathbf{w}$ and balanced photodetectors sum over all signals, producing a recovered signal $m(t)$ representing a linear combination of the received signals[19]:

$$m(t) = \mathbf{w} \cdot \mathbf{r}(t) \tag{3}$$

We seek to perform cancellation weight identification: the adjustment of $\mathbf{w}$ to match one $\mathbf{c}_i$ and thereby recover a target signal $s_i(t)$. To do so, we apply iterative implementations of principal and independent component analysis (PCA and ICA). Different photonic weight vectors $\mathbf{w}$ are tested in sequence, and the resulting $m(t)$ is sampled and its statistical properties evaluated. PCA requires maximization of the

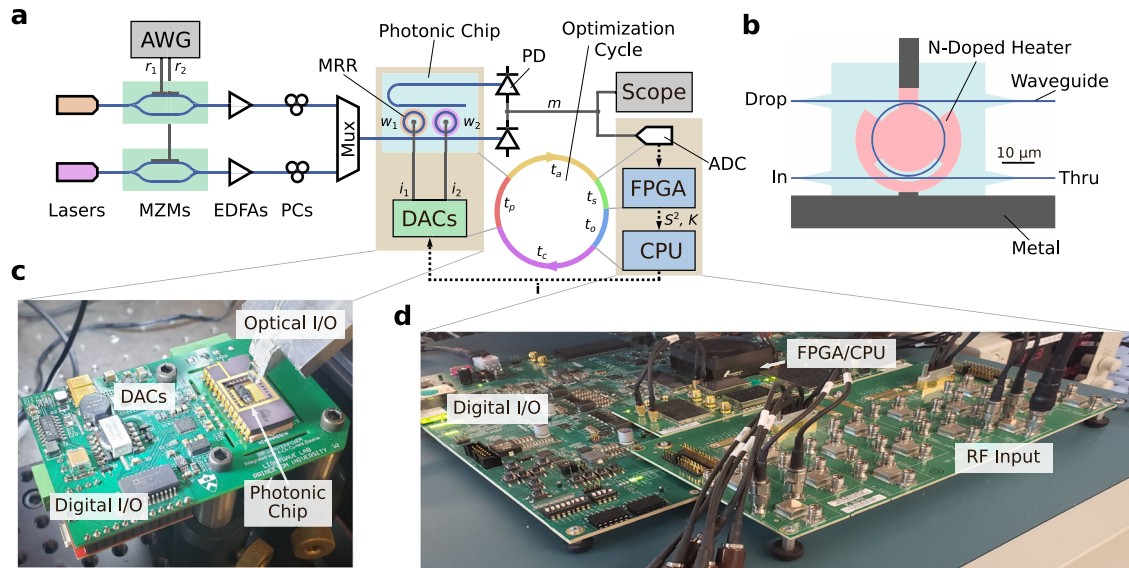

**Fig. 2 | Experimental photonic blind interference cancellation setup. a** Setup diagram. **b** Scale schematic of the experimental micro-ring resonators (variation in N-doping concentration not shown). **c** Packaged photonic chip and controller. **d** ZCU216 FPGA development board. The experimental setup generates linearly mixed signals in software and uses an arbitrary waveform generator (AWG) and Mach-Zehnder modulators (MZMs) to modulate them onto distinct wavelengths of light. An FPGA performs statistic calculation and an integrated CPU runs the optimization algorithm. EDFA: erbium-doped fiber amplifier, PC polarization controller, PD photodetector, I/O input/output.

variance of $m(t)$, $\sigma^2 = \mathbb{E}[m^2(t)]$, and ICA requires minimization of the kurtosis of $m(t)$, $\kappa = \mathbb{E}[m^4(t)]/\sigma^4 - 3$ (where it is assumed $\mathbb{E}[m(t)] = 0$ for an RF signal). Specifically, ICA relies on a property stemming from the Central Limit Theorem: when $\mathbf{w} = \mathbf{c}_i$ for some $i$—i.e., when the recovered signal matches one of the independent source signals—$\kappa$ is at a local minimum. The source signals are the independent components (ICs). PCA allows the identification of a transformed weight basis in which the cancellation weight vectors are orthogonal, ensuring they can be deterministically identified.

Experimentally, we test two representative mixing matrices with $N_r = N_s = 2$:

$$\mathbf{M}_1 = \begin{pmatrix} 0.6 & 0.4 \\ 0.4 & 0.6 \end{pmatrix} \quad \mathbf{M}_2 = \begin{pmatrix} 1 & 0.5 \\ 1 & 0.2 \end{pmatrix} \tag{4}$$

$\mathbf{M}_1$ represents a symmetrical case in which there is a similarly powerful interfering signal, corresponding to Fig. 1a. $\mathbf{M}_2$ represents a case in which powerful jamming interference masks a weaker target signal, corresponding to Fig. 1b. We use binary phase-shift keyed (BPSK) source signals with a 1 GHz carrier frequency and a 200 MBaud symbol rate.

## PBIC without Calibration

A photonic MRR weight is tuned by applying electrical current, typically to a resistive heater embedded within or near the MRR. A set of MRRs produce a weight vector $\mathbf{w}$ and are tuned by current vector $\mathbf{i}$ of the same length. While each element of $\mathbf{w}$ depends primarily on the corresponding element of $\mathbf{i}$, thermal, electrical, and optical cross-talk between the MRRs results in a current-weight transfer function best represented $\mathbf{w} = \mathbf{f}(\mathbf{i})$. This transfer function is generally modeled through calibration in order to apply accurate weights, and the sensitivity of MRRs to variations in operating temperature and optical power requires recalibration after even minute changes in either quantity[28]. While daily calibration can suffice when using a temperature- and vibration-stabilized laboratory testbed, during field operation of a PBIC system environmental stability cannot be guaranteed. Calibration is required prior to each weight identification run, and it is made complex and time-

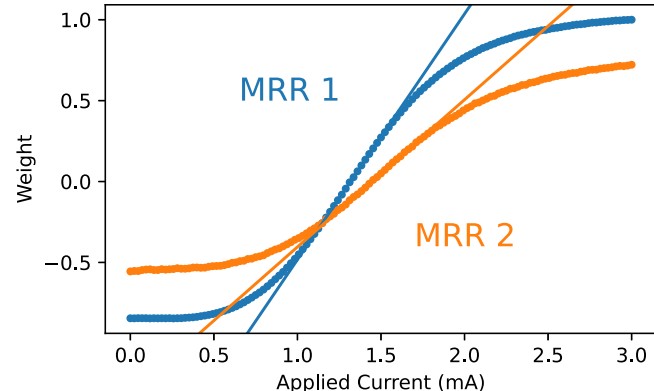

**Fig. 3 | Measured weights of micro-ring resonators (MRRs) 1 (blue) and 2 (orange) as a function of applied current.** Linear fits to the region about the zero-weight point are shown.

consuming by the need to model and compensate for multiple modes of MRR cross-talk. As the number of MRRs scale and they are placed more closely to minimize chip area, the number and strength of cross-talk interactions rises dramatically, further increasing the challenge of calibration. Nevertheless, all previous reports of useful MRR photonic systems, including systems capable of PBIC, rely on pre-calibration to determine this transfer function[23,24,27]. Alternative MRR control techniques that reduce the need for calibration require additional sensing hardware for each MRR[28,29]. Instead, we propose and demonstrate an error-robust approach to MRR control for PBIC that eliminates the need to calibrate with no additional hardware required:

We consider the shape of $\mathbf{f}$ about the zero-weight point $\mathbf{i}_0$, where $\mathbf{f}(\mathbf{i}_0) = \mathbf{0}$. Figure 3 shows the measured output weight of each MRR as its associated tuning current is swept, with the other tuning current matching $\mathbf{i}_0$. About the zero-weight point, a linear approximation of the transfer function is reasonably accurate, consistent with MRR physical modeling (see Supplementary Notes). The transfer function

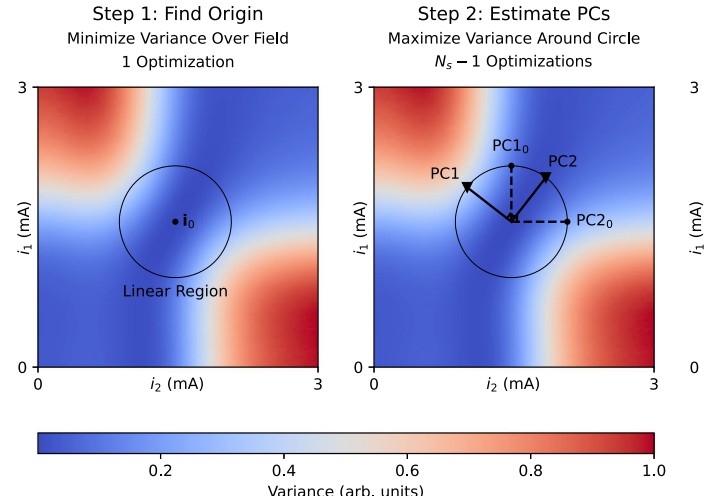

**Fig. 4 | A typical experimental photonic weight identification with the $M_2$ mixing matrix.** Step 1 identifies the point of lowest variance, $\mathbf{i}_0$. Step 2 consists of principal component analysis and step 3 of independent component analysis, each optimizing directional vectors about the circle denoting the linear region. A 0 subscript and dashed lines denote initial positions, while solid lines denote final positions. Step 4 optimizes the recovered source signals, freeing them from the linear region to reach higher-amplitude final positions denoted ′. PCs principal components, ICs independent components.

may therefore be approximated as linear near $\mathbf{i}_0$:

$$\mathbf{w} = \mathbf{f}(\mathbf{i}) \approx \left(\mathrm{D}\mathbf{f}\big|_{\mathbf{i}_0}\right)(\mathbf{i} - \mathbf{i}_0) \quad \text{given} \quad |\mathbf{i} - \mathbf{i}_0| < i_{max} \quad (5)$$

where $\mathrm{D}\mathbf{f}\big|_{\mathbf{i}_0} \in \mathbb{R}^{N_r \times N_r}$ represents the matrix of partial derivatives of $\mathbf{f}(\mathbf{i})$ evaluated at $\mathbf{i}_0$ and $i_{max}$ represents the maximum deviation from $\mathbf{i}_0$ where the linear approximation holds. Note that off-diagonal terms of $\mathrm{D}\mathbf{f}\big|_{\mathbf{i}_0}$ represent first-order approximations of MRR cross-talk that are incorporated within this model.

It follows that the recovered signal $m(t)$ may be approximated as such:

$$m(t) = \mathbf{w}^T \mathbf{M} \mathbf{s}(t) \approx \underbrace{(\mathbf{i} - \mathbf{i}_0)^T}_{\mathbf{w}'} \underbrace{\left(\mathrm{D}\mathbf{f}\big|_{\mathbf{i}_0}\right)^T \mathbf{M}}_{\mathbf{M}'} \mathbf{s}(t) \quad (6)$$

Under this approximation, there is an effective weight vector $\mathbf{w}'$, determined from the applied currents without calibration, and an effective mixing matrix $\mathbf{M}'$, the product of $\mathrm{D}\mathbf{f}\big|_{\mathbf{i}_0}$ and the true mixing matrix $\mathbf{M}$. As $\mathbf{M}'$ is unknown for PBIC, $\mathrm{D}\mathbf{f}\big|_{\mathbf{i}_0}$ does not need to be known, and only $\mathbf{i}_0$ must be identified.

We perform experimental cancellation weight identification using this approximation with $i_{max} = 0.6$ mA. Weight identification consists of the four steps shown in Fig. 4. First, variance is minimized to find $\mathbf{i}_0$, establishing the linear region as a circle of radius $i_{max}$ centered at $\mathbf{i}_0$. PCA and ICA (steps 2 and 3) operate along the edge of this circle, balancing weighting linearity with recovered signal SNR, which increases with amplitude. PCA consists of finding an orthogonal basis in which each successive basis vector points in the direction of highest variance. ICA performs an analogous process, though it operates in an adjusted basis derived from PCA and seeks to minimize kurtosis. Though the process with two signals is shown, this approach generalizes to an arbitrary number of source signals. See Supplementary Methods for a detailed discussion.

Estimates of the source signals after step 3 are subject to the cascaded errors of the previous steps stemming from noise, sampling randomness, and the linearity approximation. Furthermore, we seek to increase the amplitude of the recovered signal to maximize its SNR, but that requires pushing $\mathbf{i}$ out of the linear region. To address both problems, we add a final step to the algorithm consisting of a kurtosis minimization over the entire weight field, with each IC estimate serving

as the initial position. As noise raises the kurtosis, this step optimizes both the ICs' weights and SNRs without requiring the linearity assumption of Eq. (5). This constitutes step 4. So long as steps 1–3 place the initial IC estimates within the correct convex regions, the optimizer will find the correct kurtosis minima. Experimentally, we find that both the stronger and weaker source signals can be consistently recovered and accurately demodulated under $\mathbf{M}_1$ and $\mathbf{M}_2$.

### Low-latency adaptation

Under most wireless communication scenarios, $\mathbf{M}$ changes in response to the movement of people and objects in the environment. However, it is assumed to be static during cancellation weight identification. The desired weights must therefore be identified before $\mathbf{M}$ meaningfully changes, with the algorithm operating in real-time with low latency. Following weight identification, continuous kurtosis minimization can ensure the weight vectors remain accurate, so the latency of the initial identification represents the limiting factor. Weight identification requires a set of processing iterations, each of which consists of five sequential operations shown in Table 1. Signal acquisition refers to the collection of a set of samples of $m(t)$, the latency of which depends on the sampling rate and sample count, discussed in the following section. We demonstrate consistently successful weight identification for $\mathbf{M}_1$ and $\mathbf{M}_2$ with a signal acquisition latency down to 8.3 μs.

Statistic calculation describes the computation of the variance and kurtosis of $m(t)$ from the collected samples. In previous work, it was performed by an oscilloscope, and it dominated iteration latency. We use an alternative approach in which a Xilinx Zynq FPGA/CPU chip performs low-latency sampling, statistic calculation, and optimizer execution, as shown in Fig. 2. Our custom pipelined FPGA logic design,

### Table 1 | Contributions to Iteration Latency

| Operation | Symbol | Previous Work[27] | This Work |
|---|---|---|---|
| Signal Acquisition | $t_a$ | <1 ms | 8.3 μs (min.) |
| Statistic Calculation | $t_s$ | >1 s | 49 ns |
| Optimizer Execution | $t_o$ | <1 μs | <1 μs |
| DAC Communication | $t_c$ | 3 ms (avg.) | 3 ms (avg.) |
| Photonic Weighting | $t_p$ | 500 μs | 500 μs |
| Total | | >1 s | <4 ms |

diagrammed in Fig. 5, processes incoming 1.97 GS/s signal data in real-time, adding a negligible 49 ns latency, a greater than seven order-of-magnitude improvement.

Optimizer execution refers to the determination of the next set of weighting currents to test during iterative optimization, and it adds minimal latency. DAC communication describes the transmission of the weighting currents to the digital-to-analog converter (DAC) board. It requires an average of 3 ms, making it the primary contribution to iteration latency. Photonic weighting refers to the setting of the desired photonic weights; our system waits a conservative 500 μs to allow the weights to stabilize. We, therefore, achieve a total iteration latency below 4 ms in this work.

We implement the optimizations required for each step shown in Fig. 4 using the Nelder-Mead algorithm with a fixed 40 iterations per optimization[30]. With $N_s = 2$ experimentally, we perform five total optimizations per weight identification requiring 200 processing iterations. Total latency depends on the DAC communication and signal acquisition latency, with consistent weight identification success achievable in less than 1 s.

## Sub-Nyquist sampling

We collect $n_s$ samples of $m(t)$ at sampling rate $f_s$ to generate estimates of $\sigma^2$ and $\kappa$, denoted $S^2$ and $K$, respectively (see Methods). $n_s$ and $f_s$ dictate the uncertainty of $S^2$ and $K$. Errors in measurements of $S^2$ and $K$ during cancellation weight identification can cause an error in the estimate of an IC that leaves it outside the often small convex region with the desired kurtosis minimum, leading to weight identification failure. This motivates further analysis of the relationship between $n_s$

and $f_s$ and the weight identification success rate. Of particular interest is sub-Nyquist sampling, where $f_s$ is less than twice the signal bandwidth. Sub-Nyquist sampling has been successfully demonstrated for PBIC, but its influence on weight identification success rate has not be investigated[26,31]. There is potential for alignment between frequency components of the recovered signal and the sampling rate, generating data artifacts that increase uncertainty.

Figure 6a shows the weight identification success rate for $\mathbf{M}_1$ and $\mathbf{M}_2$ as a function of $K$ uncertainty. We find that $K$ uncertainty is strongly predictive of identification success rate, and that below a certain $K$ uncertainty threshold, which depends on the mixing matrix, successful identification becomes nearly guaranteed. The difficulty of performing weight identification for a given mixing matrix may be quantified by its ill-condition number (see Supplementary Notes). $\mathbf{M}_2$, with an ill-condition number of 7.5, represents a more challenging PBIC scenario than $\mathbf{M}_1$, with an ill-condition number of 5, accounting for the lower level of $K$ uncertainty required to reliably recover sources from $\mathbf{M}_2$ as compared to $\mathbf{M}_1$.

Figure 6b, c shows statistic uncertainty as a function of $f_s$ and $n_s$, respectively, with the other parameter held fixed. The uncertainties of $S^2$ and $K$ remain largely flat as $f_s$ varies for fixed $n_s$ (with the exception of degradation at higher sampling rates due to an experimental artifact discussed in Supplementary Notes). Our results validate that reducing $f_s$ to deeply sub-Nyquist values, an approach with significant power consumption benefits discussed in the following section, has a minimal impact on success rate. By contrast, statistic uncertainty drops sharply as $n_s$ increases, consistent with statistical theory. Weight identification success rate can be improved by raising $n_s$, but there is a trade-off.

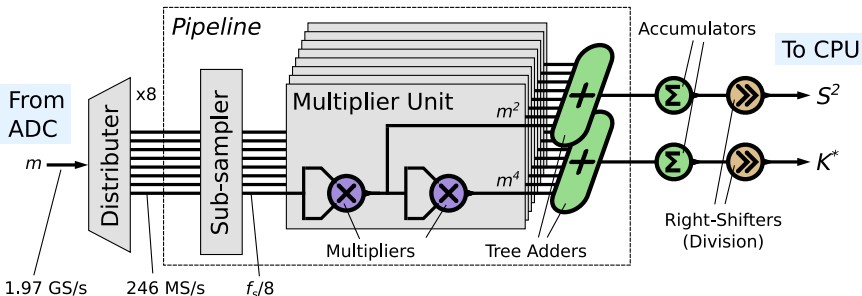

**Fig. 5 | Diagram of the FPGA-implemented statistic calculator.** Incoming samples at up to 1.97 GS/s are distributed among multiple parallel pipelined channels each capable of processing 246 MS/s, ensuring real-time processing. The FPGA implements the key operations of Eq. (10) (see Methods).

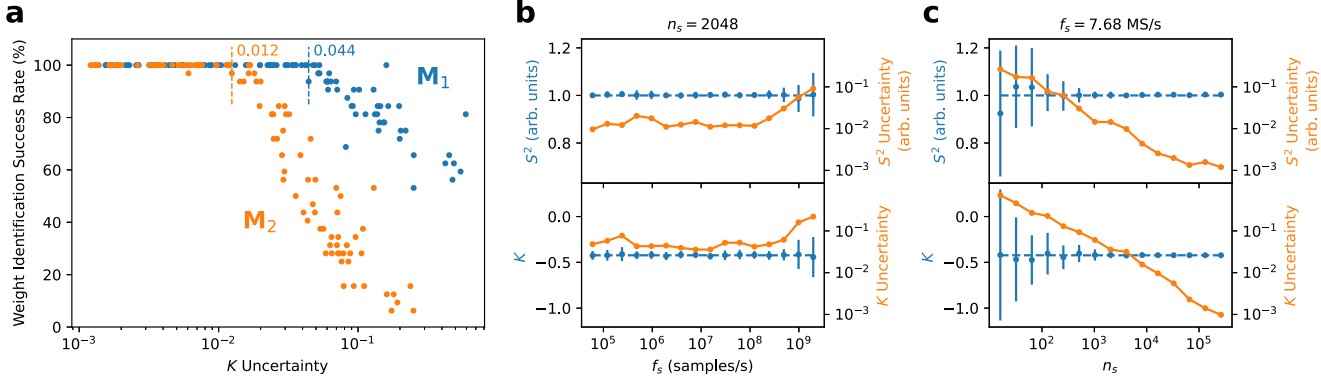

**Fig. 6 | Weight identification performance and statistic measurement uncertainty. a** Cancellation weight identification success rate as a function of $K$ uncertainty for $\mathbf{M}_1$ (blue) and $\mathbf{M}_2$ (orange). Dashed lines indicate the kurtosis uncertainty thresholds below which weight identification become near-perfectly successful, with the uncertainty values indicated. **b** Uncertainty of $S^2$ and $K$ as a function of $f_s$. **c** Uncertainty of $S^2$ and $K$ as a function of $n_s$. Blue points indicate the mean measurement of the respective statistics, with error bars indicating their uncertainty, also plotted in orange (see Methods). Dashed lines show the true underlying statistic values.

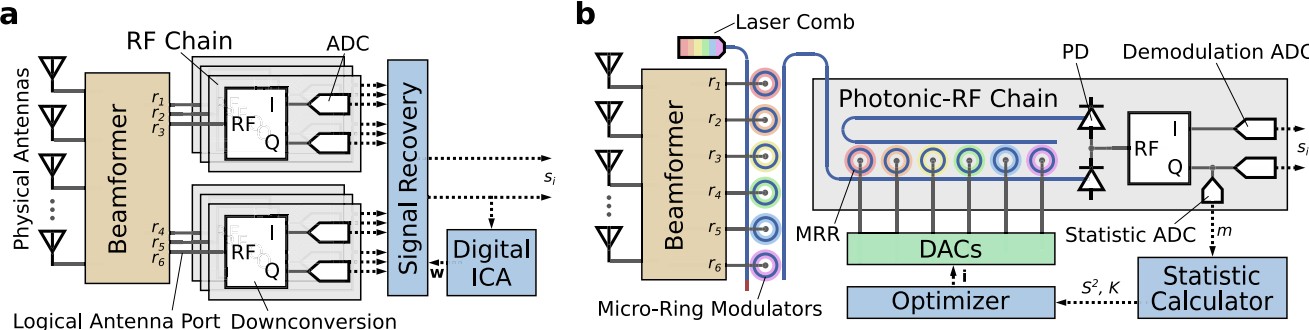

**Fig. 7 | Diagrams of hybrid beamforming receivers capable of blind interference cancellation.** Digital electronic (**a**) and analog photonic (**b**) receivers shown with $N_r = 6$ and $N_t = 1$. I in-phase, Q quadrature, ICA independent component analysis, MRR micro-ring resonator, PD photodetector.

Sample acquisition latency $t_a$, which sets the floor on system latency, depends on $f_s$ and $n_s$:

$$t_a = \frac{n_s}{f_s} \quad (7)$$

As we seek to minimize latency, we must either lower $n_s$, compromising weight identification success, or raise $f_s$, increasing power consumption. The specific sampling parameters chosen depend on the desired application. One candidate set of sampling parameters, $n_s = 2^{14}$ and $f_s = 122.9$ MS/s, results in an acquisition latency of 133 μs and a $K$ uncertainty of 0.0054, below the uncertainty threshold for both mixing matrices tested.

### A photonic-RF mmWave MIMO receiver

Building on our experimental results, we propose a hybrid digital-analog photonic beamforming receiver that performs blind interference cancellation with significantly less power consumption than the conventional digital electronic alternative. As shown in Fig. 7, in a hybrid digital-analog beamforming receiver, each physical antenna is connected by a beamforming apparatus consisting of an array of splitters, combiners, and phase shifters to a smaller number of logical antenna ports. Each logical port has an angular sensitivity that may be independently controlled via the phase-shifter array. As a result, each port provides a linear mixture of all incident source signals, with weights dependent on the beamformer tuning, the signals' angles of incidence, and the environment. These signals may be modeled by Eq. (1), allowing PBIC to be applied[4–7,15].

In a digital electronic system, the $N_r$ received signals are downconverted and digitized by $N_r$ RF chains (Fig. 7a). The analog-to-digital converters (ADCs) in each chain must operate at the Nyquist rate with bit precision sufficient to extract a weaker target signal from strong interference. After digitization, linear signal combination for signal recovery is implemented digitally. Digital ICA to determine the cancellation weights can be performed by drawing a subset of samples from the digital output signal for statistic calculation. Our ICA algorithm can be used, or alternatively the system can implement an algorithm drawn from the literature such as FastICA[32].

By contrast, in our proposed system shown in Fig. 7b, signal recovery occurs prior to downconversion and digitization in the analog photonic domain. On a single photonic chip, received signals are modulated by broadband micro-ring modulators onto spaced wavelengths produced by a co-integrated laser frequency comb[33,34]. The WDM signals are directed to $N_t$ photonic-RF chains—one for each target source—which perform recovery, downconversion, and digitization of the target source signals. The demodulation ADCs, operating at the Nyquist rate, require only the bit precision dictated by the modulation format of the target signal. Precision as low as 1 bit has been demonstrated as a way of reducing the power consumption of high-

bandwidth receivers[4,7,10,16]. Data from low-precision demodulation ADCs produce less accurate signal statistics, requiring the use of a separate, high-precision sub-Nyquist statistic ADC to drive cancellation weight identification. The weight identification itself is performed by two application-specific integrated circuits (ASICs) off the primary signal path: the statistic calculator determines $S^2$ and $K$ and the optimizer runs the weight identification algorithm.

We present a brief analysis of the power consumption of each approach to interference cancellation to motivate the use of photonics. We consider a mixing scenario with an equal number of source and received signals and one target source ($N_s = N_r$ and $N_t = 1$). The sources are quadrature phase-shift keyed (QPSK) with a symbol rate $f_Q$, and the level of interference demands signal recovery with 6 bits precision in order to extract the weaker target source. Signal statistics for the purpose of PCA and ICA are calculated at a 100 MS/s rate for both systems.

ADC power consumption can be estimated using the Schreier figure of merit ($FoM_S$), which usefully characterizes the relationship between power $P$, bit precision $b$, and sampling rate $f_s$ of a given ADC technology[35]:

$$P = \frac{f_s}{2} 10^{\frac{1}{10}(SNDR - FoM_S(f_s))} \quad (8)$$

SNDR, the signal to noise plus distortion ratio, relates to $b$ as follows:

$$SNDR = 6.02\,dB \cdot b + 1.76\,dB \quad (9)$$

As $f_s$ increases beyond 10 MS/s, $FoM_S(f_s)$ begins to fall, reflecting reduced ADC efficiency[35]. Based on recent reports, $FoM_S$ is approximately 176 dB/J at 100 MS/s[36,37], 164 dB/J at 2 GS/s[38,39], and 150 dB/J at 10 GS/s[40,41]. With these numbers, we estimate ADC power consumption.

Digital processing in both systems can be quantified in terms of multiply-accumulate (MAC) operations. For $N_t = 1$, digital signal recovery, represented by Eq. (2), requires 1 MAC for each incoming data sample. Statistic calculation requires 2 MACs per sample, visible in Fig. 5, while optimization algorithm execution requires negligible processing in comparison due to the simplicity of the Nelder-Mead algorithm. One MAC with 8-bit precision incoming values requires approximately 1 pJ of energy[42]. The rate of digital processing for digital signal recovery scales with $f_Q$. The processing for statistic calculation, however, does not, as for a given desired acquisition latency the sample count and sampling rate can remain fixed, independent of the signal symbol rate. We have shown that there is no performance penalty associated with deeply sub-Nyquist sampling.

The energy consumption of the photonic sub-system of the proposed approach is dominated by the power of the optical carriers and the micro-ring tuning current. 96 μW of optical power per GHz of

**Table 2 | Power consumption by device**

| Device | | Power (µW) | | Count | |
|---|---|---|---|---|---|
| | | $f_Q = 2$ GBaud | $f_Q = 10$ GBaud | Digital System | Photonic System |
| Nyquist ADC, 1-bit | | 0.239 | 30.0 | – | 2 |
| Nyquist ADC, 6-bit | | 244 | 30,700 | $2N_r$ | – |
| Sub-Nyquist ADC, 6-bit | | 0.771 | 0.771 | – | 1 |
| Digital Signal Recovery MAC | | 2000 | 10,000 | $2N_r$ | – |
| Statistic Calculation MAC | | 100 | 100 | 2 | 2 |
| Optical Carrier | | 192 | 960 | – | $N_r$ |
| MRR | | 120 | 120 | – | $N_r$ |
| Total | Digital System | $4,488N_r + 200$ | $81,400N_r + 200$ | | |
| | Photonic System | $312N_r + 200$ | $1,080N_r + 260$ | | |

**Table 3 | Total digitization and signal recovery power**

| $N_r$ | $f_Q$ (GBaud) | Total Power (mW) | | Improvement Factor |
|---|---|---|---|---|
| | | Digital System | Photonic System | |
| 6 | 2 | 27.1 | 2.07 | 13.1x |
| 20 | 2 | 90.0 | 6.44 | 14.0x |
| 6 | 10 | 489 | 6.74 | 72.6x |
| 20 | 10 | 1630 | 21.9 | 74.4x |

bandwidth per wavelength is required to ensure 6-bits precision given the dominant noise source at this power level, shot noise[43]. With all MRRs trimmed post-fabrication to eliminate variability[44], at most 120 µW power is required to tune the photonic weights[43].

Based on this analysis, the power consumption of each device is reported in Table 2 (see Supplementary Notes for additional details). We limit our analysis to the digitization and signal recovery elements of each system, though we note that the photonic system requires no more of any electronic RF component than the digital electronic system and significantly fewer downconverters.

Table 3 reports the total estimated digitization and signal recovery power consumption of each system under several combinations of $N_r$ and $f_Q$. The photonic system offers 14 times lower power consumption when performing interference cancellation on 2 GBaud signals and 74 times lower power consumption for 10 GBaud signals. This results from the decrease in ADC efficiency as $f_s$ increases, which affects the photonic system less due to the reduction in number of ADCs, the low bit precision of the demodulation ADCs, and the low sampling rate of the statistic ADC. We find that the photonic approach to blind interference cancellation becomes increasingly power-advantageous as $f_Q$ increases. $N_r$ has little impact on the power consumption improvement factor as all dominant power consumers in both systems scale with $N_r$.

## Discussion

In this work, we both advance PBIC technology by demonstrating techniques to address the limitations of previous work and characterize the performance of a PBIC system in order to motivate the use of photonics for interference cancellation. Our analysis indicates that PBIC can reduce signal digitization and recovery power consumption by greater than 74 times relative to the conventional digital electronic alternative, but two limitations of previous MRR-photonic systems for PBIC need to be addressed:

First, unlike digital systems MRR photonic systems are sensitive to operating temperature and optical input power. During field operation, where environmental stability is not guaranteed, complex re-calibration of the photonic system would be required prior to every weight identification run based on previous techniques. Our MRR control approach eliminates the need for calibration while still incorporating a first-order approximation of all cross-talk, allowing the updated weight identification algorithm to run immediately without concern for environmental change. Approaches like this that mitigate the thermal sensitivity of MRRs are critical to allowing any MRR

photonic system to operate without strict temperature control, and we anticipate our technique having broad application beyond RF interference cancellation.

Second, previous PBIC demonstrations faced total system latencies of several minutes, incompatible with the time-varying RF environments of the real world. We address this limitation by implementing coordinated, low-latency processing between an MRR photonic system and an FPGA, enabling real-time applied photonic weight adaptation. Our FPGA/CPU chip implements signal sampling, statistic calculation, and optimizer execution, reducing statistic calculation latency by more than seven orders of magnitude and total iteration latency by more than 200 times relative to previous work. As a result, we demonstrate sub-second total cancellation weight identification latency, consistent with real-time operation even as **M** shifts due to the movement of people and objects. We anticipate further order-of-magnitude reductions in weight identification latency with the use of low-latency digital communication protocols (e.g., serial peripheral interface) and optimization of MRR control[28]. Under this scenario, reflective of practical PBIC application, signal acquisition latency represents the primary latency contribution.

Our results show that PBIC performance has a strong dependence on recovered signal sampling rate and sample count that has not previously been characterized. These parameters determine signal acquisition latency while also impacting success rate and power consumption. We find that reducing statistic sampling rate to deeply sub-Nyquist values does not degrade PBIC success rate and offers a significant potential power consumption reduction, and we therefore conclude that it represents a crucial technique for PBIC systems, especially when operating on high-bandwidth signals. We further discover that PBIC success rate depends strongly on statistic uncertainty and thereby statistic sample count, with increased sample counts required to address more challenging signal mixing scenarios.

The power consumption benefits associated with the photonic approach to blind interference cancellation scale with received signal bandwidth. PBIC is therefore uniquely well suited to operating on high-bandwidth signals in power-constrained scenarios, though digital electronics can offer advantages in latency and technology platform maturity. As the use of high-bandwidth mmWave frequencies expands to meet increasing societal demands for wireless throughput, PBIC will become increasingly advantageous. Efficient interference cancellation can reduce the cost of developing interference-tolerant mmWave devices, enabling greater levels of multi-user spatial multiplexing and facilitating network capacity improvements[4,9].

The experimental extension of PBIC to mmWave signals will represent a key direction of future development. We have previously demonstrated PBIC up to 19.2 GHz carrier frequencies[27], and recent advancements in integrated silicon photonic components show promise toward fully extending integrated photonics to the mmWave domain[45–47]. Nevertheless, mmWave PBIC has not been shown, and an intermediate downconversion stage prior to electro-optic modulation may be required in the proposed photonic system in order to achieve low-distortion signal recovery.

## Methods

### Experimental setup

The experimental source signals are two BPSK signals consisting of distinct 1137-bit repeating random sequences. Each has a 200 MBaud data rate and a carrier frequency offset from 1 GHz by 176 kHz in opposite directions to prevent artifacts generated from a perfect alignment (the frequency offset is not used for signal discrimination). The signals are mixed in software and generated by a Keysight N8196A 92 GS/s AWG, which modulates the signals using MZMs onto distinct C-band laser frequencies generated by two Pure Photonics PPCL500 lasers. The light is polarization-controlled, amplified, and coupled onto a photonic chip, which performs signal recovery. The output intensities are received by a Discovery Semiconductor DSC-R405ER balanced photodetector, and the resulting signal is split between a Tektronix DPO73304SX 100 GS/s oscilloscope and an analog input to the Xilinx ZCU216 FPGA development board. The FPGA board communicates over a serial protocol with a custom printed circuit board (PCB) that includes DACs to apply the weighting currents (Fig. 2c). The photonic chip is electrically connected to a chip carrier on the PCB using wire-bonds.

### Digital processing setup

All digital processing is performed by the ZCU216 board using a Xilinx XCZU49DR chip with a cointegrated ADC, FPGA, and Arm cortex. The ADC samples the measured signal continuously at a fixed rate, feeding data to the FPGA logic fabric on which the statistic calculator shown in Fig. 5 is implemented. All FPGA logic is clocked at 246 MHz, and samples, arriving at up to 1.97 GS/s, are therefore distributed between 8 identical parallel paths. Lower sampling rates are implemented by dropping samples as appropriate. Each processing path calculates the second and fourth power of each sample, their contributions to the variance and kurtosis, respectively, assuming the signal mean is zero. These contributions are then accumulated from among the paths and over the course of the calculation. As the contributions from each sample are independent, the processing can be efficiently pipelined, with calculation occurring in parallel with signal sampling. Once all samples have been processed, the accumulated sums are divided by the number of samples using right-shifting, as the number of samples is required to be a power of 2. The result is $S^2$, the variance estimate, and $K^*$, a preliminary value used to determine the kurtosis estimate as follows: $K = K^*/S^4 - 3$. This final operation is performed on a floating-point basis by the Arm cortex. Xilinx High-Level Synthesis (HLS) was used to create this logic design.

Total end-to-end processing by the FPGA requires 12 clock cycles, corresponding to an additional latency (over the signal acquisition latency) of 49 ns. The Nelder-Mead optimization algorithm is executed by the Arm cortex. The algorithm interfaces with the DAC control board over a serial protocol to set weights.

### Statistic calculation

$n_s$ samples of $m(t)$, denoted $m_1, m_2, \ldots, m_{n_s}$, are used to generate estimators $S^2$ and $K$ of the variance and kurtosis, respectively:

$$S^2 = \frac{1}{n_s}\sum_{i=1}^{n_s} m_i^2 \quad K = \frac{1}{S^4}\frac{1}{n_s}\sum_{i=1}^{n_s} m_i^4 - 3 \tag{10}$$

The quality of these estimators for the purpose of PBIC is quantified by their uncertainty. The uncertainty of an estimator $A$ is equal to its standard deviation $\sigma(A)$:

$$\sigma(A) = \sqrt{\mathbb{E}\left[(A - \mathbb{E}[A])^2\right]} \tag{11}$$

### Data collection

For a given mixing matrix and set of sampling parameters, we estimate the uncertainty of a statistic estimator by taking standard deviation of 32 consecutive estimator measurements taken at the high-variance $\mathbf{i} = (0\,\text{mA}, 3\,\text{mA})$ point.

Data shown in Fig. 6a were collected under a fixed set of sampling rates $f_s$ and sample counts $n_s$. $f_s$ varied from 960 kS/s to 1.97 GS/s by power-of-2 scaling factors, and $n_s$ varied from $2^8$ to $2^{16}$ by powers of 2. Cancellation weight identification success rate is defined as the percentage of weight identification attempts, out of 32, which enable the successful demodulation of both source signals with no bit errors over the full bit sequence. Tests on all combinations of allowed $f_s$ and $n_s$ values are shown. $S^2$ and $K$ uncertainty data shown in Fig. 6b, c was collected with the $\mathbf{M}_1$ mixing matrix.

## Data availability

The PBIC data generated in this study have been deposited in the Figshare database under accession code https://doi.org/10.6084/m9.figshare.24556474.

## Code availability

All code used in this study is available from the corresponding author upon request.

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

## Acknowledgements

This research is supported by the National Science Foundation (NSF) (ECCS-2128616 and ECCS-1642962 to P.P.), the Office of Naval Research (ONR) (N00014-18-1-2297, N00014-20-1-2664, and N00014-22-1-2527 to P.P.), and the Defense Advanced Research Projects Agency (HR00111990049 to P.P.). The devices were fabricated at the Advanced Micro Foundry (AMF) in Singapore through the support of CMC Microsystems. J.C.L. acknowledges support from the Department of Defense (DoD) through the National Defense Science & Engineering Graduate (NDSEG) Fellowship Program. B.J.S. acknowledges support from the Natural Sciences and Engineering Research Council of Canada (NSERC). S.B. acknowledges funding from the Fonds de recherche du Québec—Nature et technologies.

## Author contributions

J.C.L., W.Z., and T.F.L. conceived the idea for the experiment. J.C.L. conceived the application of PBIC to MIMO interference cancellation and developed the proposed design and the zero-calibration MRR control scheme. W.Z. developed the experimental photonic setup, including the DAC control board and the associated control software. J.C.L. integrated the ZCU216 and wrote the associated software, including the FPGA fabric design and the implementation of the weight identification algorithm, with support from T.F.L. and W.Z. T.F.L., W.Z., S.B., and E.C.B. provided theoretical and experimental support. J.C.L. wrote the manuscript with support from T.F.L., B.J.S., and W.Z. P.P. supervised the research and contributed to the vision and execution of the experiment.

## Competing interests

The authors declare no competing interests.
