## [Peer Review File · Nature Communications]

Real-Time Photonic Blind Interference CancellationREVIEWER COMMENTS

Reviewer #1 (Remarks to the Author):

In this manuscript, a photonic mmWave MIMO receiver architecture is presented for interference cancellation. Real-time photonic interference cancellation is achieved by using zero-calibration micro-ring resonator control algorithm in an integrated FPGA photonic system. The innovative part is that interference cancellation is realized using MRRS, which can perform vector multiplication. I have some comments about this work.

- 1) In my opinion, there is no need to compare the proposed photonic cancellation with the analog cancellation. They are two different technique roadmaps. Each one has its own advantages and disadvantages.
- 2) The authors claimed that this technique has reduced space and power needs, thus is suitable for handheld mobile devices. However, FPGA and CPU are needed to carry out the algorithm, which are not low power consumption. What is the size and power of the photonic chip?
- 3) 200 MBaud data rate and the carrier frequency is 1 GHz. What if faster and higher carrier frequency is used?
- 4) What is the cost and complexity of this method?
- 5) The proposed work is a beamforming receiver, which has only two photonic-RF channels. Does it mean that the channel capacity is only doubled? As shown in Fig, 2, if we need six channels in parallel, should we use six photonic-RF chains?
- 6) The novelty should be emphasized. Many techniques used in this method have been proposed in referenced designs. They are combined to design this system.
- 7) How to connect the photonic chip with beamformer?
- 8) The PBSS success rate is stable only when the SNR is higher than 20 dB. However, in practical environment, the SNR might be low. The effectiveness of the proposed PBSS should be explained.
- 9) The M matrix only has one interfering signal or jamming. In practical environment, there should be multiple interfering signals. Is it still usable?
- 10) The latency of the whole system is estimated to be 30 ms. How to obtain this value? Can it be measured?

Reviewer #2 (Remarks to the Author):

the authors present a cutting edge approach to addressing inference in MIMO RF communication links. Their offer resides that inference cancellation problem is done in the analogue domain, instead of replicating a DSP chain for each RF antenna. Their key argument is that similar to [21] blind source separation using photonic MRR components is beneficial compared to RF analogue electronics. From my understanding the key difference of this manuscript compared to [21] is that processing latency due to the inclusion has been radically reduced, rendering the scheme, application "ready". Another difference is a more robust way to control optical weights without pre-calibration.

The manuscript is well written and contains interesting and sound results.

On the other hand, one of main concerns is that compared to [21] the authors main advantage is the generation of a more application ready system, where statistics for variance and kurtosis are computed by a fast FPGA thus latency is minimised. In this context, this work although really interesting seems incremental and thus not so fit for an article in Nature Communications. In this context, I think it would be beneficial if the authors could include:

- a more in-depth discussion regarding the issue of using analogue RF processing? Does the use of additional laser sources and modulators per antenna outweigh these issues and if yes can it be quantified?

- Regarding zero-calibration a quantification regarding the latency reduction compared to the standard case would be beneficial.

- During MRR-zero calibration if crosstalk between the MRRs is present how it will affect convergence of the system and what is the minimum crosstalk that allow efficient operation of the system

- A minor comment the authors mention that "For properly chosen optical wavelengths, the photonic transfer function f , where $w = f(i)$, may be approximated as linear about the zero-weight point i_0 " can the authors present more details on that taken into consideration that it is a critical part of the manuscript.

Reviewer #3 (Remarks to the Author):

This manuscript proposes that they develop real-time photonic interference cancellation with an integrated FPGA photonic system that executes a novel zero-calibration micro-ring resonator control algorithm. Multiple-input multiple-output (MIMO) mmWave devices are hot topics because they can increase the transfer rate efficiently. The description and explanation of the whole article are rather straightforward. However, I still have several questions about this article. Detailed comments are listed below.

1. In this article, it has been mentioned many times that the new system has such low latency and anti-interference improvement compared with the traditional system. However, there is no actual data in comparison. Perhaps the data supplemented by the two comparative comparisons is more intuitive.

2. In Figure 2a, I noticed that for each logical antenna port, an RF chain is needed to digitize, and after ADC processes each three ports, the signal of each three ports will be processed as a source signal. Can you describe this DSP process in detail?

3. Figure 5 shows PBSS performance and statistic estimator consistency under two conditions, m_1 , and m_2 (corresponding to fig1a and fig1b, respectively). We do better when we're dealing with symmetrical cases in which there is a similarly powerful interfering signal. Can you explain why?

4. This paper presents a practical interference cancellation system and its advantages over traditional systems. As a practical engineering problem, besides the benefits, it is evident that there will inevitably be some unsatisfactory places compared with the conventional system. Can you talk about the defects?

Response to Reviewers' Comments

Reviewer 1

In this manuscript, a photonic mmWave MIMO receiver architecture is presented for interference cancellation. Real-time photonic interference cancellation is achieved by using zero-calibration micro-ring resonator control algorithm in an integrated FPGA photonic system. The innovative part is that interference cancellation is realized using MRRS, which can perform vector multiplication. I have some comments about this work.

1) *In my opinion, there is no need to compare the proposed photonic cancellation with the analog cancellation. They are two different technique roadmaps. Each one has its own advantages and disadvantages.*

Response: We agree that analog photonic cancellation and digital electronic cancellation have very different behaviors and performances, which result in each one being best suited to different domains. In the original manuscript the motivation for including the comparison of the two approaches was not sufficiently clear, and we neglected to analyze the advantages and drawbacks photonics offers to blind interference cancellation.

Changes: The approach comparison has been moved to *Results Section D*. We have added an analysis of the power consumption of the two approaches to that section to motivate the use of photonics. Tables II and III (see below) document our analysis results, which indicate that photonic interference cancellation offers a substantial power consumption reduction.

2) *The authors claimed that this technique has reduced space and power needs, thus is suitable for handheld mobile devices. However, FPGA and CPU are needed to carry out the algorithm, which are not low power consumption. What is the size and power of the photonic chip?*

Response: Yes, we use a Zynq FPGA/CPU chip experimentally to implement digital processing, and these contribute non-negligibly to the power consumption of the experimental system. However, we use only a small portion of the processing resources of the chip, and simple ASICs are capable of fully implementing our algorithm without the overhead of an FPGA or CPU. For this reason, we propose using ASICs in a practical system, though designing them is infeasible in an experimental context. This was not sufficiently clear in the original manuscript.

Photonic chips consist of micron-scale components which combine to form photonic systems with total dimensions on the order of 100s of microns, though further miniaturization is possible. These dimensions are consistent with inclusion in handheld devices.

We appreciate that the reviewer points out that we neglect to analyze power consumption in the original manuscript.

Changes: We have moved discussion of the proposed system to *Results Section D* to better distinguish between it and our experimental setup.

We have removed references to the potential size reduction of our proposed system. While we believe the reduction in the number of RF components within the system will result in a size reduction, the power consumption reduction our proposed system offers represents the primary motivation for its use.

We have added an analysis of power consumption of digital electronic and analog photonic approaches to blind interference cancellation to *Results Section D*. Our analysis incorporates estimates of the power consumption associated with digital processing in both systems and of the photonic components in the photonic systems. Table II and Table III document our results, and we find that the photonic system offers a substantial digitization and signal recovery power consumption reduction:

TABLE II
POWER CONSUMPTION BY DEVICE

Device	Power (μ W)		Count	
	$f_Q = 2$ GBaud	$f_Q = 10$ GBaud	Digital System	Photonic System
Nyquist ADC, 1-bit	0.239	30.0	–	2
Nyquist ADC, 6-bit	244	30,700	$2N_r$	–
Sub-Nyquist ADC, 6-bit	0.771	0.771	–	1
Digital Signal Recovery MAC	2,000	10,000	$2N_r$	–
Statistic Calculation MAC	100	100	2	2
Optical Carrier	192	960	–	N_r
MRR	120	120	–	N_r
Total				
Digital System	$4,488N_r + 200$	$81,400N_r + 200$		
Photonic System	$312N_r + 200$	$1,080N_r + 260$		

TABLE III
TOTAL DIGITIZATION AND SIGNAL RECOVERY POWER

N_r	f_Q (GBaud)	Total Power (mW)		Improvement Factor
		Digital System	Photonic System	
6	2	27.1	2.07	13.1x
20	2	90.0	6.44	14.0x
6	10	489	6.74	72.6x
20	10	1,630	21.9	74.4x

3) 200 MBaud data rate and the carrier frequency is 1 GHz. What if faster and higher carrier frequency is used?

Response: We have demonstrated successful photonic blind interference cancellation up to 19.2 GHz using the same setup that we use in our work. We chose to use a 1 GHz carrier frequency in this work in order to validate our ability to operate in real-time and collect performance data, and we believe that there are no fundamental challenges in extending the techniques we demonstrate up to the 19.2 GHz level. For the purpose of statistic calculation behavior, operating on higher carrier frequencies is analogous to sampling at a slower rate, which we demonstrate causes no performance degradation in our system.

In order to operate at the mmWave scale beyond 30 GHz, changes to the experimental setup may be required. We suspect an intermediate downconversion step applied to the received signals prior to electro-optic modulation may be necessary to ensure the photonic devices we use to perform interference cancellation operate as we expect them to. Evaluating the changes that need to be made to operate experimentally on mmWave signals will be a valuable direction for future work.

Changes: We make the following changes and additions to discuss the scaling of our approach to higher data rates and carrier frequencies:

We add a note that the statistic processing rate does not change with signal bandwidth f_Q :

Results Section D: “The rate of digital processing for digital signal recovery scales with f_Q . The processing for statistic calculation, however, does not, as for a given desired acquisition latency the sample count and sampling rate can remain fixed, independent of the signal symbol rate. We have shown that there is no performance penalty associated with deeply sub-Nyquist sampling.”

We add a paragraph considering of the extension of this work to higher carrier frequencies and data rates:

Discussion: “The experimental extension of PBIC to mmWave signals will represent a key direction of future development. We have previously demonstrated PBIC up to 19.2 GHz carrier frequencies, and recent advancements in integrated silicon photonic components show promise toward fully extending integrated photonics to the mmWave domain. Nevertheless, mmWave PBIC has not been shown, and an intermediate downconversion stage prior to electro-optic modulation may be required in the proposed photonic system in order to achieve low-distortion signal recovery.”

4) *What is the cost and complexity of this method?*

Response: The primary costs of the proposed photonic approach as compared to the conventional digital one are associated with fabrication of the photonic chip. Silicon photonic chip fabrication represents a newer technology than traditional digital and RF electronic chip fabrication, but it has become increasingly mature in recent years.

Digital interference cancellation offers excellent latency characteristics for blind interference cancellation. We have sought, in this paper, to address limitations of photonics in this area, but nevertheless photonic-specific contributions to latency may impose an overall latency penalty in comparison to a digital implementation.

Our analysis indicates that photonics can enable a substantial reduction in power consumption as compared to digital electronics.

Changes: We have added Table I to better quantify the latency associated with a photonic implementation of blind interference cancellation. *Results Section B* includes additional details on latency.

TABLE I
CONTRIBUTIONS TO ITERATION LATENCY

Operation	Symbol	Previous Work [27]	This Work
Signal Acquisition	t_a	< 1 ms	8.3 μ s (min.)
Statistic Calculation	t_s	> 1 s	49 ns
Optimizer Execution	t_o	< 1 μ s	< 1 μ s
DAC Communication	t_c	3 ms (avg.)	3 ms (avg.)
Photonic Weighting	t_p	500 μ s	500 μ s
Total		> 1 s	< 4 ms

In *Results Section D*, we have added an analysis of the power consumption of the conventional digital electronic and our analog photonic approaches to blind interference cancellation. We estimate that photonics can enable a substantial power consumption reduction.

5) The proposed work is a beamforming receiver, which has only two photonic-RF channels. Does it mean that the channel capacity is only doubled? As shown in Fig. 2, if we need six channels in parallel, should we use six photonic-RF chains?

Response: The capacity of a MIMO system depends on *both* the number of spatial channels and the data throughput of each spatial channel. RF interference can reduce the throughput of each channel, and our system aims to mitigate interference in order to increase the throughput per channel and thereby the total throughput. We don't aim to increase the number of channels available.

We assume that there are strong interference sources in the environment which are disrupting the receiver's spatial channels. In a conventional digital system, those interference sources would have to be isolated and digitized in order to allow the digital system to subtract them from the received signals and recover the target signals. This digitization requires significant power consumption, and it is unnecessary from a fundamental perspective as we are uninterested in the data the interfering sources are transmitting.

Our system performs signal recovery in the analog domain, eliminating the need to digitize one received signal for each source signal. For example, if there are two target source signals and four interference source signals, a conventional system would have to have six RF chains to perform interference cancellation while our system would only require two photonic-RF chains (as shown in Fig. 2 in the original manuscript). If all six sources were target sources and there were no other sources of interference, we would need six photonic-RF chains, but that is not the condition our proposed system is optimized for.

Changes: We have added details clarifying key design parameters of the system: the number of source signals, the number of received signals, and the number of target signals:

Results: "We consider a set of N_s spatially-separated transmitters communicating with a single receiver. There are N_s independent source signals ... These source signals mix over the air, such that the receiver detects a set of N_r distinct linear mixtures of those signals ... where we require $N_r \geq N_s$."

"Blind interference cancellation describes the task of recovering, from the received signals, a subsection of the source signals numbering N_t , the *target* source signals, where $N_t \leq N_s$."

We have adjusted Fig. 7, formerly Fig. 2, to show one photonic-RF chain, reflecting a prototypical system, with the caption explicitly referencing these parameters:

Fig. 7 Caption: "Digital electronic (a) and analog photonic (b) mmWave hybrid beamforming receivers capable of blind interference cancellation with $N_r = 6$ and $N_t = 1$."

We have also made explicit the number of RF chains and photonic-RF chains required:

Results Section D: “In a digital electronic system, the N_r received signals are downconverted and digitized by N_r RF chains”

“[In the analog photonic system t]he WDM signals are directed to N_t photonic-RF chains—one for each target source—which perform recovery, downconversion, and digitization of the target source signals”

6) *The novelty should be emphasized. Many techniques used in this method have been proposed in referenced designs. They are combined to design this system.*

Response: We appreciate that the reviewer points out that in the original manuscript we do not sufficiently distinguish between previously reported approaches and our own novel contributions. We believe we report three primary novel experimental results:

- A successful zero-calibration approach to photonic micro-ring resonator control
- A low-latency, real-time implementation of photonic blind interference cancellation taking advantage of FPGA coprocessing
- Characterization of the relationship between the system's statistic sample rate and sample count and system latency, power consumption, and success rate

We further propose a photonic mmWave receiver capable of blind interference cancellation with greatly reduced signal recovery and digitization power consumption as compared to the conventional digital electronic alternative.

Changes: To make our novel contributions clear to readers, we have reorganized the manuscript, dividing our results between four subsections, each corresponding to one of the contributions listed above. The last subsection, focusing on our proposed system, includes a new analysis of the power consumption of blind interference cancellation systems in order to motivate the use of photonics. The summary of our results in the introduction has been rewritten to clearly document previous work as well as the limitations of that work which we address in our work:

Introduction: “[W]e have recently demonstrated photonic blind interference cancellation (PBIC) with 9 bits of weight precision and effective control of signals from DC to 19.2 GHz. However, like all previous MRR photonic systems, that PBIC system required re-calibration after any small shift in operating temperature or optical input power, and identifying the correct cancellation weights required minutes, incompatible with real-time operation.

“In this work, we demonstrate a PBIC system with key advancements that address limitations observed in prior work. Our system relies on a novel zero-calibration approach to MRR control for PBIC that greatly reduces the complexity of adapting to changes in temperature and optical power. Digital processing in our system is implemented on a Xilinx Zynq chip containing a field-programmable gate array (FPGA) and a central processing unit (CPU), resulting in a greater than 200-fold reduction in latency and sub-second cancellation weight identification latency. We demonstrate low-latency coordinated processing between an MRR photonic system and an FPGA as well as real-time applied photonic weight adaptation, both novel to the authors' knowledge. Our results highlight that the statistic sampling rate and sample count represent critical parameters impacting the latency, power consumption, and success rate of a PBIC system. We establish that sub-Nyquist sampling is a crucial technique for reducing power consumption without compromising PBIC success. Finally, we propose a novel mmWave beamforming receiver architecture capable of PBIC and estimate that it can achieve a greater than 74-fold reduction in digitization and signal recovery power in comparison to the conventional digital electronic alternative.”

1) *How to connect the photonic chip with beamformer?*

Response: Experimentally, the photonic chip is connected to a chip-carrier on our custom photonic controller PCB using wire-bonds. In our proposed system, all optical devices, including lasers, are co-integrated on a single photonic chip, ensuring only electrical interfacing is required. Standard integrated circuit packaging techniques can be applied to create a surface-mount photonic blind interference cancellation chip. RF connections between the photonic chip and the beamformer can therefore be implemented using RF-optimized PCB lines.

Changes: We have added the following to clarify how we experimentally implement electrical interfacing with the photonic chip:

Methods Section A: “The photonic chip is electrically connected to a chip carrier on the PCB using wire-bonds.”

2) The PBSS success rate is stable only when the SNR is higher than 20 dB. However, in practical environment, the SNR might be low. The effectiveness of the proposed PBSS should be explained.

Response: We confusingly use the term SNR to refer to two distinct measurements in the original manuscript: the SNR of the received signals from the environment (the conventional usage) and the SNR of our calculated statistics (which depends on our sampling parameters). The statistic SNR is critical in determining the effectiveness of the system, and it can be improved by increasing sample count regardless of the received signal SNR. In this way our system can address low received-signal SNR environments. This “statistic SNR” terminology is not standard, and we thank the reviewer for pointing out this lack of clarity.

Changes: We have removed the term statistic SNR, instead quantifying the potential variability of our statistics in terms of a statistic uncertainty, the standard terminology. Fig. 6, formerly Fig. 5, has been regenerated using statistical uncertainty, and the terminology has been changed throughout the manuscript.

Figure 6:

Methods Section C:

[Original Manuscript] “The quality of these estimators for the purpose of PBSS is quantified by their SNR. The SNR of an estimator A is defined as the ratio of the estimator mean $\mu(A)$... to the estimator standard deviation $\sigma(A)$ ”

[Revised Manuscript] “The quality of these estimators for the purpose of PBIC is quantified by their uncertainty. The uncertainty of an estimator A is equal to its standard deviation $\sigma(A)$ ”

3) *The M matrix only has one interfering signal or jamming. In practical environment, there should be multiple interfering signals. Is it still usable?*

Response: Additional interfering signals makes the signal recovery process more challenging, but we believe our system remains effective. Our approach is built on PCA and ICA, which generalize to arbitrarily large mixing matrices—we discuss the complete algorithm in Supplementary Methods. For the purpose of initial experimental validation, we chose to use the simplest case of two source and two received signals. We hope, in future work, to extend our results in larger numbers of interfering signals and to characterize performance as a function of number of interfering signals.

Changes: We have added a clarification of the generality of our approach:

Results Section A: “Though the process with two signals is shown, this approach generalizes to an arbitrary number of source signals. See Supplementary Methods for a detailed discussion.”

4) The latency of the whole system is estimated to be 30 ms. How to obtain this value? Can it be measured?

Response: Weight identification in the experimental system required 5 optimizations (a consequence of operating on 2 received signals). Each optimization consists of a series of weight-set/statistic measurement iterations. We chose to use 40 iterations for each optimization, as we found that ensured consistent convergence. A total of 200 iterations were therefore required to perform complete weight identification.

There are five operations in each iteration: signal acquisition, statistic calculation, optimizer execution, DAC communication, and photonic weighting. In the original manuscript we estimated that all operations aside from signal acquisition could be considered negligible in our proposed system with sufficient optimization. One set of sampling parameters resulting in a high success rate corresponded to a 133 μ s acquisition latency. We therefore concluded that a total latency of 30 ms, slightly above 200 times the acquisition latency to account for other latency sources, is achievable conservatively based on our results. We appreciate the reviewer for pointing out that our calculation methodology was not clear in the original manuscript.

Changes: We have added Table I, explicitly listing each of the five operations in each iteration as well as their associated latency in previous work and in this work. In this work average iteration latency fell below 4 ms, corresponding to a total experimental weight identification latency of less than one second. Iteration latency extended longer than one second in previous work, resulting in a total weight identification latency in the minutes.

TABLE I
CONTRIBUTIONS TO ITERATION LATENCY

Operation	Symbol	Previous Work [27]	This Work
Signal Acquisition	t_a	< 1 ms	8.3 μ s (min.)
Statistic Calculation	t_s	> 1 s	49 ns
Optimizer Execution	t_o	< 1 μ s	< 1 μ s
DAC Communication	t_c	3 ms (avg.)	3 ms (avg.)
Photonic Weighting	t_p	500 μ s	500 μ s
Total		> 1 s	< 4 ms

Results Section B has been adapted to discuss each of the five operations in greater detail.

Fig. 4 has been updated to explicitly document the number of optimizations per step in the weight identification algorithm:

We have added a paragraph documenting how total weight identification latency is calculated:

Results Section B: “We implement the optimizations required for each step shown in Fig. 4 using the Nelder-Mead algorithm with a fixed 40 iterations per optimization. With $N_s = 2$ experimentally, we perform five total optimizations per weight identification requiring 200 processing iterations. Total latency depends on the DAC communication and signal acquisition latency, with consistent weight identification success achievable in less than one second.”

We have removed references to the 30 ms value from the manuscript, as we conclude that the origin of that value is not sufficiently clear. Our primary experimental latency results are sub-second total weight identification latency and a greater than 200-fold improvement in iteration latency.

Reviewer 2

The authors present a cutting edge approach to addressing inference in MIMO RF communication links. Their offer resides that inference cancellation problem is done in the analogue domain, instead of replicating a DSP chain for each RF antenna. Their key argument is that similar to [21] blind source separation using photonic MRR components is beneficial compared to RF analogue electronics. From my understanding the key difference of this manuscript compared to [21] is that processing latency due to the inclusion has been radically reduced, rendering the scheme, application "ready". Another difference is a more robust way to control optical weights without pre-calibration. The manuscript is well written and contains interesting and sound results.

On the other hand, one of main concerns is that compared to [21] the authors main advantage is the generation of a more application ready system, where statistics for variance and kurtosis are computed by a fast FPGA thus latency is minimised. In this context, this work although really interesting seems incremental and thus not so fit for an article in Nature Communications.

Response: We appreciate the reviewer's perspective, and we agree that our paper is motivated by the results of [21], but we believe the primary breakthroughs of our manuscript center on areas of fundamental innovation not reported in [21]. These are:

- A zero-calibration approach to micro-ring resonator control (as the reviewer acknowledges)
- A low-latency, real-time implementation of photonic blind interference cancellation using coordinated processing between an FPGA and an MRR photonic chip
- Experimental characterization of the relationship between statistic sampling parameters and system performance, validating sub-Nyquist sampling for blind interference cancellation

We further propose a photonic mmWave receiver capable of blind interference cancellation with greatly reduced signal recovery and digitization power consumption as compared to the conventional digital electronic alternative.

We did not make our areas of innovation sufficiently clear in our original manuscript, and we did not sufficiently distinguish between the results of previous work and the results of this work.

Changes: We have substantially reorganized and revised the manuscript to make our areas of innovation clear.

Our summary of our results in the Introduction has been rewritten to clearly reflect the limitations of previous work that we address in this work:

Introduction: “[W]e have recently demonstrated photonic blind interference cancellation (PBIC) with 9 bits of weight precision and effective control of signals from DC to 19.2 GHz. However, like all previous MRR photonic systems, that PBIC system required re-calibration after any small shift in operating temperature or optical input power, and identifying the correct cancellation weights required minutes, incompatible with real-time operation.

“In this work, we demonstrate a PBIC system with key advancements that address limitations observed in prior work. Our system relies on a novel zero-calibration approach to MRR control for PBIC that greatly reduces the complexity of adapting to changes in temperature and optical power. Digital processing in our system is implemented on a Xilinx Zynq chip containing a field-programmable gate array (FPGA) and a central processing unit (CPU), resulting in a greater than 200-fold reduction in latency and sub-second cancellation weight identification latency. We demonstrate low-latency

coordinated processing between an MRR photonic system and an FPGA as well as real-time applied photonic weight adaptation, both novel to the authors' knowledge. Our results highlight that the statistic sampling rate and sample count represent critical parameters impacting the latency, power consumption, and success rate of a PBIC system. We establish that sub-Nyquist sampling is a crucial technique for reducing power consumption without compromising PBIC success. Finally, we propose a novel mmWave beamforming receiver architecture capable of PBIC and estimate that it can achieve a greater than 74-fold reduction in digitization and signal recovery power in comparison to the conventional digital electronic alternative.”

The Results section has been reorganized into four subsections, each corresponding to one of the contributions we list above, so that our novel contributions are more clear. These are:

- PBIC without Calibration
- Low-Latency Adaptation
- Sub-Nyquist Sampling
- A Photonic-RF mmWave MIMO Receiver

We have added a number of additional details to better document the advancement of our approach and the value of photonics to blind interference cancellation:

- A quantitative documentation of latency relative to previous work (Table I)
- A diagram of the custom FPGA logic design that enabled real-time statistic calculation (Fig. 5)
- A quantitative power consumption analysis of a photonic system in comparison to the digital electronic alternative (*Results Section D*)

Our Discussion has been rewritten to clearly convey the impact of our advancements on photonic blind interference cancellation specifically and MRR photonic systems more broadly:

Discussion: “In this work, we both advance PBIC technology by demonstrating novel techniques to address the limitations of previous work and characterize the performance of a PBIC system in order to motivate the use of photonics for interference cancellation. Our analysis indicates that PBIC can reduce signal digitization and recovery power consumption by greater than 74 times relative to the conventional digital electronic alternative, but two limitations of previous MRR-photonic systems for PBIC needed to be addressed:

“First, MRR photonic systems are sensitive to operating temperature and optical input power. During field operation, where environmental stability is not guaranteed, complex re-calibration of the photonic system would be required prior to every weight identification run based on previous techniques. Our novel MRR control approach *eliminates* the need for calibration while still incorporating a first-order approximation of all cross-talk, allowing the updated weight identification algorithm to run immediately without concern for environmental change. Approaches like this that mitigate the thermal sensitivity of MRRs are critical to allowing any MRR photonic system to operate without strict temperature control, and we anticipate our technique having broad application beyond RF interference cancellation.

“Second, previous PBIC demonstrations faced total system latencies of several minutes, incompatible with the time-varying RF environments of the real world. We addressed this limitation by implementing, for the first time, coordinated, low-latency processing between an MRR photonic system and an FPGA, enabling real-time applied photonic weight adaptation. Our FPGA/CPU chip implements signal sampling, statistic calculation, and optimizer execution, reducing statistic calculation latency by more than seven orders of magnitude and total iteration latency by more than 200 times

relative to previous work. As a result, we demonstrate sub-second total cancellation weight identification latency, consistent with real-time operation even as M shifts due to the movement of people and objects. We anticipate further order-of-magnitude reductions in weight identification latency with the use of low-latency digital communication protocols (e.g. serial peripheral interface) and optimization of MRR control. Under this scenario, reflective of practical PBIC application, signal acquisition latency represents the primary latency contribution.

“Our results show that PBIC performance has a strong dependence on recovered signal sampling rate and sample count that has not previously been characterized. These parameters determine signal acquisition latency while also impacting success rate and power consumption. We find that reducing statistic sampling rate to deeply sub-Nyquist values does not degrade PBIC success rate and offers a significant potential power consumption reduction, and we therefore conclude that it represents a crucial technique for PBIC systems, especially when operating on high-bandwidth signals. We further discover that PBIC success rate depends strongly on statistic uncertainty and thereby statistic sample count, with increased sample counts required to address more challenging signal mixing scenarios.”

In this context, I think it would be beneficial if the authors could include

1. a more in-depth discussion regarding the issue of using analogue RF processing? Does the use of additional laser sources and modulators per antenna outweigh these issues and if yes can it be quantified?

Response: We unfortunately neglect to thoroughly analyze the advantage of our analog photonic approach in the original manuscript. An analog photonic blind cancellation system offers an order-of-magnitude power consumption reduction as compared to a digital electronic system, resulting primarily from the reduction in the number, sampling rate, and sampling precision of the ADCs required. While the photonic components do add meaningfully to the power consumption, on net the power consumption is vastly reduced under the photonic approach.

We believe photonic signal recovery offers benefits over analog electronic signal recovery relating to higher bandwidths, lower loss, and resistance to electro-magnetic interference. Photonics also offers efficient signal interconnection using wavelength-division multiplexing. These capabilities make photonics well suited to operation with high-bandwidth spatially-multiplexed protocols that enable high data throughputs.

Changes: We have adjusted the introduction to clarify the advantages of photonics that motivate its use for blind interference cancellation:

Introduction: “The RF photonics platform, where RF signals are modulated onto optical carriers and processed in the optical domain, offers high bandwidths, low loss, and resistance to electro-magnetic interference. A single integrated silicon photonic waveguide can carry dozens of high-bandwidth RF signals each on a separate wavelength, enabling hardware-efficient interconnection.”

We have also added a substantial discussion of the power consumption of digital electronic and analog photonic components to *Results Section D*. We conclude that a photonic blind interference canceller can require greater than 74-fold less power for signal recovery and digitization as compared to the digital electronic alternative. Tables II and III document the power consumption results:

TABLE II
POWER CONSUMPTION BY DEVICE

Device	Power (μ W)		Count	
	$f_Q = 2$ GBaud	$f_Q = 10$ GBaud	Digital System	Photonic System
Nyquist ADC, 1-bit	0.239	30.0	–	2
Nyquist ADC, 6-bit	244	30,700	$2N_r$	–
Sub-Nyquist ADC, 6-bit	0.771	0.771	–	1
Digital Signal Recovery MAC	2,000	10,000	$2N_r$	–
Statistic Calculation MAC	100	100	2	2
Optical Carrier	192	960	–	N_r
MRR	120	120	–	N_r
Total				
Digital System	$4,488N_r + 200$	$81,400N_r + 200$		
Photonic System	$312N_r + 200$	$1,080N_r + 260$		

TABLE III
TOTAL DIGITIZATION AND SIGNAL RECOVERY POWER

N_r	f_Q (GBaud)	Total Power (mW)		Improvement Factor
		Digital System	Photonic System	
6	2	27.1	2.07	13.1x
20	2	90.0	6.44	14.0x
6	10	489	6.74	72.6x
20	10	1,630	21.9	74.4x

2. Regarding zero-calibration a quantification regarding the latency reduction compared to the standard case would be beneficial.

Response: Calibration is typically performed by sweeping the tuning current of each micro-ring resonator individually while measuring the output intensity. We have found success in the past using approximately 16 points per MRR covering the full weight sweep. Our experimental system in this work uses two MRRs, so naive calibration under this model requires a total of 32 points. Each point requires one weight-set/statistic measurement iteration.

Table I, in the revised manuscript, reports the latency per iteration, with additional details provided in *Results Section B* (see below). Less than 4 ms are required per iteration in this work. We can therefore estimate the latency associated with experimental calibration to be no more than 128 ms. This value is small in comparison to other latency sources in the experimental system.

The primary benefit of our zero-calibration approach relates to the challenges posed by MRR cross-talk and the way in which those challenges scale with number of MRRs. Thermal, electrical, and optical cross-talk represent major barriers to effective calibration and control of MRRs, especially as the number of MRRs scales. We anticipate that a number of additional weight-sweeps would be required to estimate cross-talk, and there are significant challenges relating to modeling and control complexity.

Changes: We have clarified the motivation of the zero-calibration algorithm:

Introduction: "Our system relies on a novel zero-calibration approach to MRR control for PBIC that greatly reduces the complexity of adapting to changes in temperature and optical power."

We have added Table I, which quantifies latency in our system:

TABLE I
CONTRIBUTIONS TO ITERATION LATENCY

Operation	Symbol	Previous Work [27]	This Work
Signal Acquisition	t_a	< 1 ms	8.3 μ s (min.)
Statistic Calculation	t_s	> 1 s	49 ns
Optimizer Execution	t_o	< 1 μ s	< 1 μ s
DAC Communication	t_c	3 ms (avg.)	3 ms (avg.)
Photonic Weighting	t_p	500 μ s	500 μ s
Total		> 1 s	< 4 ms

We have added detail on the calculation of total latency from iteration latency:

Results Section B: "We implement the optimizations required for each step shown in Fig. 4 using the Nelder-Mead algorithm with a fixed 40 iterations per optimization. With $N_s = 2$ experimentally, we perform five total optimizations per weight identification requiring 200 processing iterations. Total latency depends on the DAC communication and signal acquisition latency, with consistent weight identification success achievable in less than one second."

We add greater discussion of cross-talk and its implications for our system. As the reviewer's next concern covers cross-talk, these changes are noted on the next page.

3. *During MRR-zero calibration if crosstalk between the MRRs is present how it will affect convergence of the system and what is the minimum crosstalk that allow efficient operation of the system*

Response: MRR cross-talk has three modes: thermal cross-talk between one MRR's heater and an adjacent MRR, optical cross-talk as one MRR's resonance curve affects adjacent wavelengths, and electrical cross-talk associated with parasitic resistance. We design our MRR's to ensure the effect of cross-talk on a given photonic weight is small in comparison to the effect of its tuning current. Thermal cross-talk, typically the dominant form of cross-talk, can be proportionally reduced by ensuring a MRR heater is as close as possible to the MRR. We use N-doped heaters formed of the same silicon that creates the optical waveguide, allowing us to heat the waveguide directly. For this reason, we assume that the cross-talk, while meaningful, is small in comparison to the primary tuning effect.

We model the electro-optic transfer function as linear in the region around the zero-weight point. We observe that this represents a reasonable approximation of the primary tuning effect: the relationship between an MRR's weight and its heater current. Cross-talk is also approximated to the first degree in our model, represented by off-diagonal terms of the matrix of partial derivatives of $f(\mathbf{i})$. As the underlying mechanism of action of cross-talk is the same as that of the primary tuning effect, as the cross-talk effect is much smaller than that of the primary tuning effect, and as the primary tuning effect can be reasonably approximated linearly, we believe the linear approximation of cross-talk incorporated in our model is quite accurate.

For this reason, we conclude that any physically reasonable cross-talk in a well-designed broadcast-and-weight photonic blind interference cancellation system such as ours will not have a deleterious effect on system performance. Cross-talk is incorporated in the system model, and it therefore is compensated implicitly during PCA and ICA.

Changes: We add details to better convey the challenges associated with MRR calibration in order to motivate a zero-calibration approach:

Results Section A: “A photonic MRR weight is tuned by applying electrical current, typically to a resistive heater embedded within or near the MRR. A set of MRRs produce a weight vector \mathbf{w} and are tuned by current vector \mathbf{i} of the same length. While each element of \mathbf{w} depends primarily on the corresponding element of \mathbf{i} , thermal, electrical, and optical cross-talk between the MRRs result in a current-weight transfer function best represented $\mathbf{w} = f(\mathbf{i})$. This transfer function is generally modeled through calibration in order to apply accurate weights, and the sensitivity of MRRs to variations in operating temperature and optical power requires recalibration after even minute changes in either quantity. While daily calibration can suffice when using a temperature- and vibration-stabilized laboratory testbed, during field operation of a PBIC system environmental stability cannot be guaranteed. Calibration is required prior to each weight identification run, and it is made complex and time-consuming by the need to model and compensate for multiple modes of MRR cross-talk. As the number of MRRs scale and they are placed more closely on a chip to minimize chip area, the number and strength of cross-talk interactions rises dramatically, further increasing the challenge of calibration. Nevertheless, all previous reports of useful MRR photonic systems, including systems capable of PBIC, rely on pre-calibration to determine this transfer function. Alternative MRR control techniques that reduce the need for calibration require additional sensing hardware for each MRR.”

We note that a first-order approximation of cross-talk is included in our zero-calibration model:

Results Section A: “Note that off-diagonal terms of Df_{i0} represent first-order approximations of MRR cross-talk that are incorporated within this model.”

Discussion: “Our novel MRR control approach *eliminates* the need for calibration while still incorporating a first-order approximation of all cross-talk, allowing the updated weight identification algorithm to run immediately without concern for environmental change.”

4. A minor comment the authors mention that "For properly chosen optical wavelengths, the photonic transfer function f , where $w = f(i)$, may be approximated as linear about the zero-weight point i_0 " can the authors present more details on that taken into consideration that it is a critical part of the manuscript.

Response: We appreciate that the reviewer points out that we have not sufficiently justified this claim. The linearity of the electro-optic transfer function about the zero-weight point is an experimental observation, supported by physical modeling. In Supplementary Notes, we provide a detailed analysis of a physical model of the transfer function that shows that a high-linearity point can exist at the zero-weight operating point. Our experimental observations support this conclusion, though we neglected to include them in the original manuscript. We additionally note that errors in the linear approximation can be corrected during Step 4 of the weight identification algorithm, providing additional justification for the usefulness of the approximation.

Changes: We have added Fig. 3 to the manuscript showing the experimental electro-optic transfer function of each of the two MRRs used in this work. We include a linear model of the transfer function about the zero-weight point in order to support the reasonableness of the linear approximation. Though the approximation loses accuracy as the target weight increases from zero, it nevertheless is sufficient to identify reasonable IC estimates:

Fig. 3:

Results Section A: "Fig. 3 shows the measured output weight of each MRR as its associated tuning current is swept, with the other tuning current matching i_0 . About the zero-weight point, a linear approximation of the transfer function is reasonably accurate, consistent with MRR physical modeling (see Supplementary Notes). The transfer function may therefore be approximated as linear near i_0 ."

Reviewer 3:

This manuscript proposes that they develop real-time photonic interference cancellation with an integrated FPGA photonic system that executes a novel zero-calibration micro-ring resonator control algorithm. Multiple-input multiple-output (MIMO) mmWave devices are hot topics because they can increase the transfer rate efficiently. The description and explanation of the whole article are rather straightforward. However, I still have several questions about this article. Detailed comments are listed below.

- 1. In this article, it has been mentioned many times that the new system has such low latency and anti-interference improvement compared with the traditional system. However, there is no actual data in comparison. Perhaps the data supplemented by the two comparative comparisons is more intuitive.*

Response: Conventional digital cancellation systems have excellent latency and interference cancellation characteristics, and we don't aim to improve on them in this work. Rather, our goal is to address the high power consumption associated with digital cancellation. Digitization at high data rates tends to be power-hungry, and analog cancellation prior to digitization can reduce both the number of ADCs required and the digitization precision necessary. This is our motivation for investigating analog photonic interference cancellation.

We focus on latency in our work because that was a major limitation of previous analog photonic interference cancellation work. Low-latency, real-time operation is a requirement for useful application of this technology, and the FPGA integration in our experimental system is key to achieving it. The need for regular calibration was also a limitation of previous work that we address in this work.

Changes: We have rewritten the end of our Introduction to clarify the motivation of our work and the novel contributions we offer:

Introduction: "Analog interference cancellation, where interference is subtracted prior to signal digitization, can reduce system power consumption. We propose to implement it using RF photonics.

...

"In particular, we have recently demonstrated photonic blind interference cancellation (PBIC) with 9 bits of weight precision and effective control of signals from DC to 19.2 GHz. That PBIC system, like all previous MRR photonic systems, required re-calibration after any small shift in operating temperature or optical input power, and identification of the correct cancellation weights required minutes, incompatible with real-time operation.

"In this work, we demonstrate a PBIC system with key advancements that address the limitations of previous work. Our system relies on a novel zero-calibration approach to MRR control for PBIC that greatly reduces the complexity of adapting to changes in temperature and optical power. Digital processing in our system is implemented on an FPGA/CPU chip, resulting in a greater than 200-fold latency reduction and sub-second cancellation weight identification latency. We demonstrate low-latency coordinated processing between an MRR photonic system and an FPGA as well as real-time applied photonic weight adaptation, both novel to the authors' knowledge. Based on our results, statistic sampling rate and sample count represent key parameters impacting the latency, power consumption, and success rate of a PBIC system, and we establish that sub-Nyquist sampling represents a critical technique for reducing power consumption without compromising PBIC success. Finally, we propose a

novel mmWave beamforming receiver architecture capable of PBIC and estimate that it offers a greater than 74-fold digitization and signal recovery power reduction in comparison to the conventional digital electronic alternative."

We have added Table I, which quantifies the latency advantage our system offers relative to previous work. We reduce weight-set/statistic measurement iteration latency from greater than 1 s to less than 4 ms, a greater than 200-fold improvement. *Results Section B* includes substantial additional latency details.

TABLE I
CONTRIBUTIONS TO ITERATION LATENCY

Operation	Symbol	Previous Work [27]	This Work
Signal Acquisition	t_a	< 1 ms	8.3 μ s (min.)
Statistic Calculation	t_s	> 1 s	49 ns
Optimizer Execution	t_o	< 1 μ s	< 1 μ s
DAC Communication	t_c	3 ms (avg.)	3 ms (avg.)
Photonic Weighting	t_p	500 μ s	500 μ s
Total		> 1 s	< 4 ms

We have added a new analysis of the power consumption of the analog photonic and the digital electronic approaches to *Results Section D*. Tables II and III document our results, with photonics offering a substantial digitization and signal recovery power reduction:

TABLE II
POWER CONSUMPTION BY DEVICE

Device	Power (μ W)		Count	
	$f_Q = 2$ GBaud	$f_Q = 10$ GBaud	Digital System	Photonic System
Nyquist ADC, 1-bit	0.239	30.0	–	2
Nyquist ADC, 6-bit	244	30,700	$2N_r$	–
Sub-Nyquist ADC, 6-bit	0.771	0.771	–	1
Digital Signal Recovery MAC	2,000	10,000	$2N_r$	–
Statistic Calculation MAC	100	100	2	2
Optical Carrier	192	960	–	N_r
MRR	120	120	–	N_r
Total				
	Digital System	$4,488N_r + 200$	$81,400N_r + 200$	
	Photonic System	$312N_r + 200$	$1,080N_r + 260$	

TABLE III
TOTAL DIGITIZATION AND SIGNAL RECOVERY POWER

N_r	f_Q (GBaud)	Total Power (mW)		Improvement Factor
		Digital System	Photonic System	
6	2	27.1	2.07	13.1x
20	2	90.0	6.44	14.0x
6	10	489	6.74	72.6x
20	10	1,630	21.9	74.4x

2. In Figure 2a, I noticed that for each logical antenna port, an RF chain is needed to digitize, and after ADC processes each three ports, the signal of each three ports will be processed as a source signal. Can you describe this DSP process in detail?

Response: The digital signal processing performed by the digital system (formerly Fig. 2a, now Fig. 7a) has two parts. First, signal recovery, represented by Eq. 3 of the revised manuscript, must be performed on a continuous basis in order to cancel interference and recover the target source signal(s). Second, ICA must be performed in the digital domain in order to identify the correct cancellation weights using one of a variety of algorithms in the literature, such as FastICA.

Changes: We have adjusted the referenced figure to add details on digital processing. Digital processing is split between "Signal Recovery" and "Digital ICA."

Fig. 7:

The digital processing is documented as follows:

Results Section D: "After digitization, linear signal combination for signal recovery is implemented digitally. Digital ICA to determine the cancellation weights can be performed by drawing a subset of samples from the digital output signal for statistic calculation. Our ICA algorithm can be used, or alternatively the system can implement an algorithm drawn from the literature such as FastICA."

3. *Figure 5 shows PBSS performance and statistic estimator consistency under two conditions, m_1 , and m_2 (corresponding to fig1a and fig1b, respectively). We do better when we're dealing with symmetrical cases in which there is a similarly powerful interfering signal. Can you explain why?*

Response: The difficulty in performing signal recovery in a given mixing scenario may be quantified by the mixing matrix's ill-condition number, which indicates how sensitive the recovered output is to errors in the cancellation weights. When operating in a scenario with a high mixing matrix ill-condition number, a small error in the weight may throw off the recovered signal enough to cause the weight identification to fail, while the same error in a low ill-condition number scenario may result in success. The ill-condition number is a unitless value equal to the product of the L_2 norm of the matrix and its inverse. \mathbf{M}_1 has an ill-condition number of 5, while \mathbf{M}_2 has a higher number of 7.5, accounting for the increased challenge in demixing \mathbf{M}_2 .

Changes: We have made the following addition to the manuscript detailing how the ill-condition numbers of the mixing matrices can explain their different behavior:

Results Section C: "The difficulty of performing weight identification for a given mixing matrix may be quantified by its ill-condition number (see Supplementary Notes). \mathbf{M}_2 , with an ill-condition number of 7.5, represents a more challenging PBIC scenario than \mathbf{M}_1 , with an ill-condition number of 5, accounting for the lower level of K uncertainty required to reliably recover sources from \mathbf{M}_2 as compared to \mathbf{M}_1 ."

We have also added a section entitled "Matrix Ill-Condition Numbers" to Supplementary Notes to provide additional details on the calculation of the ill-condition numbers.

4. *This paper presents a practical interference cancellation system and its advantages over traditional systems. As a practical engineering problem, besides the benefits, it is evident that there will inevitably be some unsatisfactory places compared with the conventional system. Can you talk about the defects?*

Response:

Analog photonic interference has a few potential areas of weakness in comparison to the digital electronic alternative:

- Digital electronics has excellent latency characteristics, which analog photonics has not yet been shown to match
- MRR cross-talk must be compensated in photonic systems
- Silicon photonic interference cancellation has not yet been demonstrated up to mmWave carrier frequencies
- Digital systems are, by nature, resistant to noise, while analog systems are susceptible to it
- Digital electronics represents a mature technology platform, while silicon photonics is still maturing
- The on-chip integrated laser comb in the proposed system is not yet generally available

In this work we seek to address several of these weaknesses, improving the latency of photonic blind interference cancellation, making it more robust to errors, and better addressing cross-talk. Other challenges are being addressed in industry or in other research laboratories.

We believe the power-consumption benefits of photonics outweigh any weaknesses of the platform under many scenarios, particularly when operating with high levels of interference or with high-bandwidth signals.

Changes: We consider limitations of the photonic implementation of blind interference cancellation in additional content at several places in the revised manuscript:

Introduction: "[L]ike all previous MRR photonic systems, that PBIC system required re-calibration after any small shift in operating temperature or optical input power, and identifying the correct cancellation weights required minutes, incompatible with real-time operation."

Results Section A: "[The electro-optic] transfer function is generally modeled through calibration in order to apply accurate weights, and the sensitivity of MRRs to variations in operating temperature and optical power requires recalibration after even minute changes in either quantity. While daily calibration can suffice when using a temperature- and vibration-stabilized laboratory testbed, during field operation of a PBIC system environmental stability cannot be guaranteed. Calibration is required prior to each weight identification run, and it is made complex and time-consuming by the need to model and compensate for multiple modes of MRR cross-talk."

Discussion: "The experimental extension of PBIC to mmWave signals will represent a key direction of future development. We have previously demonstrated PBIC up to 19.2 GHz carrier frequencies, and recent advancements in integrated silicon photonic components show promise toward fully extending integrated photonics to the mmWave domain. Nevertheless, mmWave PBIC has not been shown, and an intermediate downconversion stage prior to electro-optic modulation may be required in the proposed photonic system in order to achieve low-distortion signal recovery."

REVIEWER COMMENTS

Reviewer #1 (Remarks to the Author):

All my comments have been well addressed.

Reviewer #2 (Remarks to the Author):

The authors have thoroughly addressed all the issues raised. The revised manuscript is clearer and more detailed thus it is fit for publication

Reviewer #3 (Remarks to the Author):

I proposed four questions affecting the article's interpretability and integrity during the last review. I'm glad the author gave most of the questions an excellent revision. Based on the revised paper, I will provide detailed responses to each of the questions I asked last time.

1. In this article, it has been mentioned many times that compared with the traditional system, the new system has such low latency and anti-interference improvement, but I did not find the actual data comparison, perhaps the data supplemented by the two comparative comparisons is more intuitive.

I can see a direct comparison between this system and previous work, both from the abstract and the content. The author provides us with two tables that show the latency and power consumption improvement and the detailed methods of how they get the results. I believe this problem has been solved perfectly.

2. In Figure 2a, I noticed that for each logical antenna port, an RF chain is needed to digitize, and after each of the three ports is processed by ADC, the signal of each of the three ports will be processed as a source signal. Can you describe this DSP process in detail?

The original Figure 2 is now moved to Figure 7, and I just noticed that the original Figure 2a was the conventional hybrid beamforming receiver. In that case, the DSP process details are not that important. But now I can see minor changes in Figure 7a, which replaces DSP with signal recovery + Digital ICA, which is undoubtedly better.

3. Figure 5 shows PBSS performance and statistic estimator consistency under two conditions, m1 and m2 (corresponding to fig1a and fig1b, respectively). We do better when we're dealing with symmetrical cases in which there is a similarly powerful interfering signal. Can you explain why? Now, the interpretation of the data in the figure is more detailed, and the reason for the curve is basically clarified.

4. This paper presents a practical interference cancellation system and also shows its advantages over traditional systems. As a practical engineering problem, in addition to the advantages, it is obvious that compared with the traditional system, there will inevitably be some unsatisfactory places. Can you talk about the defects?

I didn't see any comments added for this question. However, this part is just a little bit of my personal confusion. But I do understand that it's common to highlight the strengths and hide the weaknesses in an article. This is not a big issue.

All in all, I believe the revision of this article is successful, especially in response to the question that this article lacks intuitive data to support the advantage of the system. I noticed that the author provided sufficient details about the data about the methods and how they calculate it, which is excellent.

Response to Reviewers' Comments

Reviewers 1 and 2 have indicated that all of their comments have been addressed and that they are satisfied with the revised draft. Reviewer 3 returned four questions in response to the original draft. In the reviewer's response to the revised draft, the reviewer indicated that questions 1 and 2 have been fully satisfied. This minor revision aims to address any remaining concerns relating to questions 3 and 4 of Reviewer 3.

Reviewer 3

3.

[Original Comments] *Figure 5 shows PBSS performance and statistic estimator consistency under two conditions, m_1 , and m_2 (corresponding to fig1a and fig1b, respectively). We do better when we're dealing with symmetrical cases in which there is a similarly powerful interfering signal. Can you explain why?*

[Comments on Revision] *Now, the interpretation of the data in the figure is more detailed, and the reason for the curve is basically clarified.*

Response:

In our original revision, we made the following changes to provide an explanation for the difference in performance between the two mixing matrices:

Results Section C: "The difficulty of performing weight identification for a given mixing matrix may be quantified by its ill-condition number (see Supplementary Notes). M_2 , with an ill-condition number of 7.5, represents a more challenging PBIC scenario than M_1 , with an ill-condition number of 5, accounting for the lower level of K uncertainty required to reliably recover sources from M_2 as compared to M_1 ."

We have also added a section entitled "Matrix Ill-Condition Numbers" to Supplementary Notes to provide additional details on the calculation of the ill-condition numbers.

The challenge level of each matrix may be quantified by its ill-condition number. As M_2 has a higher ill-condition number than M_1 , performance during demixing for the second mixing matrix is worse. We believe the original revision fully addresses the reviewer's question on how to account for the difference in performance.

4.

[Original Comments] *This paper presents a practical interference cancellation system and also shows its advantages over traditional systems. As a practical engineering problem, in addition to the advantages, it is obvious that compared with the traditional system, there will inevitably be some unsatisfactory places. Can you talk about the defects?*

[Comments on Revision] *I didn't see any comments added for this question. However, this part is just a little bit of my personal confusion. But I do understand that it's common to highlight the strengths and hide the weaknesses in an article. This is not a big issue.*

Response: In our original response and revision to this question, we provided the following:

Response: Analog photonic interference has a few potential areas of weakness in comparison to the digital electronic alternative:

- Digital electronics has excellent latency characteristics, which analog photonics has not yet been shown to match
- MRR cross-talk must be compensated in photonic systems
- Silicon photonic interference cancellation has not yet been demonstrated up to mmWave carrier frequencies
- Digital systems are, by nature, resistant to noise, while analog systems are susceptible to it
- Digital electronics represents a mature technology platform, while silicon photonics is still maturing
- The on-chip integrated laser comb in the proposed system is not yet generally available

In this work we seek to address several of these weaknesses, improving the latency of photonic blind interference cancellation, making it more robust to errors, and better addressing cross-talk. Other challenges are being addressed in industry or in other research laboratories.

We believe the power-consumption benefits of photonics outweigh any weaknesses of the platform under many scenarios, particularly when operating with high levels of interference or with high-bandwidth signals.

Changes: We consider limitations of the photonic implementation of blind interference cancellation in additional content at several places in the revised manuscript:

Introduction: "[L]ike all previous MRR photonic systems, that PBIC system required recalibration after any small shift in operating temperature or optical input power, and identifying the correct cancellation weights required minutes, incompatible with real-time operation."

Results Section A: "[The electro-optic] transfer function is generally modeled through calibration in order to apply accurate weights, and the sensitivity of MRRs to variations in operating temperature and optical power requires recalibration after even minute changes in either quantity. While daily calibration can suffice when using a temperature- and vibration-stabilized laboratory testbed, during field operation of a PBIC system environmental stability cannot be guaranteed. Calibration is required prior to each weight identification run, and it is made complex and time-consuming by the need to model and compensate for multiple modes of MRR cross-talk."

Discussion: "The experimental extension of PBIC to mmWave signals will represent a key direction of future development. We have previously demonstrated PBIC up to 19.2 GHz carrier frequencies, and recent advancements in integrated silicon photonic components show promise toward fully extending integrated photonics to the mmWave domain. Nevertheless, mmWave PBIC has not been shown, and an intermediate downconversion stage prior to electro-optic modulation may be required in the proposed photonic system in order to achieve low-distortion signal recovery."

Here, we highlight the following three areas as the most important weaknesses of a photonic system like the one we demonstrate as compared to the conventional digital approach:

- Greater sensitivity to operating temperature, requiring frequent calibration with consideration of cross-talk
- Higher latency
- Less mature technological platform

The challenges associated with calibration and cross-talk are discussed in detail in the revised manuscript (see above).

Changes: We add the following to clarify these limitations:

Discussion: "[U]nlike digital systems MRR photonic systems are sensitive to operating temperature and optical input power."

"PBIC is therefore uniquely well suited to operating on high-bandwidth signals in power-constrained scenarios, though digital electronics can offer advantages in latency and technology platform maturity."

REVIEWERS' COMMENTS

Reviewer #3 (Remarks to the Author):

The authors have addressed the comments in the revised manuscript satisfactorily.